# Wound Healing Activity of α-Pinene and α-Phellandrene

**DOI:** 10.3390/molecules26092488

**Published:** 2021-04-24

**Authors:** Judith Salas-Oropeza, Manuel Jimenez-Estrada, Armando Perez-Torres, Andres Eliu Castell-Rodriguez, Rodolfo Becerril-Millan, Marco Aurelio Rodriguez-Monroy, Katia Jarquin-Yañez, Maria Margarita Canales-Martinez

**Affiliations:** 1Laboratorio de Farmacognosia, UBIPRO Facultad de Estudios Superiores Iztacala, UNAM, 54108 Tlalnepantla, Mexico; judithsalazo@hotmail.com (J.S.-O.); rbecerrilm85@gmail.com (R.B.-M.); 2Instituto de Quimica, UNAM, Circuito Exterior, Ciudad Universitaria, 02860 D.F., Mexico; manueljemex@gmail.com; 3Facultad de Medicina, UNAM, Circuito Exterior, Ciudad Universitaria, 02860 D.F., Mexico; armandop@unam.mx (A.P.-T.); castell@unam.mx (A.E.C.-R.); katys12@hotmail.com (K.J.-Y.); 4Carrera de Medicina Facultad de Estudios Superiores-Iztacala, UNAM, 54108 Tlalnepantla, Mexico; dr.marcorodriguezmonroy@gmail.com

**Keywords:** terpenes, *Bursera*, essential oils, wound healing

## Abstract

*Bursera morelensis* is used in Mexican folk medicine to treat wounds on the skin. Recently, it was shown that the essential oil (EO) of *B. morelensis* has wound healing activity, accelerating cutaneous wound closure and generating scars with good tensile strength. α-pinene (PIN) and α-phellandrene (FEL) are terpenes that have been found in this EO, and it has been shown in different studies that both have anti-inflammatory activity. The aim of this study was to determine the wound healing activity of these two terpenes. The results of in vitro tests demonstrate that PIN and FEL are not cytotoxic at low concentrations and that they do not stimulate fibroblast cell proliferation. In vivo tests showed that the terpenes produce stress-resistant scars and accelerate wound contraction, due to collagen deposition from the early stages, in wounds treated with both terpenes. Therefore, we conclude that both α-pinene and α-phellandrene promote the healing process; this confirms the healing activity of the EO of *B. morelensis*, since having these terpenes as part of its chemical composition explains part of its demonstrated activity.

## 1. Introduction

Essential oils (EOs) are complex volatile natural compounds consisting of terpenoid hydrocarbons, oxygenated terpenes, and sesquiterpenes. They are liquid and volatile lipophilic substances characterized by strong aromas. EOs are synthesized by aromatic plants through their secondary metabolism [1]. Their characteristics have made them highly valued by industry, for use in food, cosmetic, and pharmaceutical applications. These natural products are potent antioxidants, free radical scavengers, and metal chelators; preclinical studies have reported the antinociceptive, neuroprotective, anticonvulsant, and anti-inflammatory effects of EOs, so they are regarded as potential sources for the development of new drugs [2,3,4,5].

Several studies have shown that a system of channels in the cortex of genus *Bursera* trees produces an EO with a high concentration of volatile terpenoids [6,7,8,9,10,11]. *B. morelensis* is a tree endemic to Mexico that has been traditionally used for the treatment of skin wounds. The people of San Rafael, Coxcatlan (Puebla, Mexico), make a tea with the bark of this species to wash wounds; it has been verified that the EO acts as an anti-inflammatory compound with antimicrobial and antifungal activity [12,13,14,15,16]. Our research team has recently shown that the EO of *B. morelensis* has wound healing activity, accelerating the speed of cutaneous wound closure and generating scars with good tensile strength [17].

Using individual chemical compounds instead of essential oils allows one to test the functions of individual chemicals and delve into the mechanisms of action of both essential oils and the pure compounds [18]. Monoterpenes are the most important constituents of essential oils produced through liquid extraction and steam distillation of edible and medicinal plants [19]. Monoterpenes are C_10_, plastid derived terpenoids, often with appreciable volatility [20]. Among the terpenes that have been found in the EO of *B. morelensis* are α-pinene (PIN) and α-phellandrene (FEL) [15,17], particularly in our previous work where it was shown that the EO of *B. morelensis* has healing activity and we found PIN in a concentration very close to 9% and FEL in a concentration very close to 1% [17] so, in this work, it was decided to test the healing activity of these terpenes at these concentrations. Both terpenes show anti-inflammatory activity [21,22,23]. PIN significantly suppresses the production of cyclooxygenase 2 (COX-2) [21], which is essential for inflammation, but it has also been shown to inhibit tumor necrosis factor alpha (TNF-α), interleukin 1 beta (IL-1β), and interleukin 6 (IL-6) during acute pancreatitis [21]. Moreover, PIN has been reported to have antiproliferative activity via inducing cell cycle arrest and apoptotic cell death in different ways [24,25]. It has been found that α-phellandrene (FEL) inhibits the production of proinflammatory cytokines such as TNF-α and IL-6 [23], and has antinociceptive effects [26]. Furthermore, FEL exhibits anticancer activity by inducing cell cycle arrest and apoptosis [27].

Although there is growing interest in understanding the mechanisms underlying monoterpenes’ pharmacological activity [28], little is known about the biological effects of PIN and FEL. Considering that these monoterpenes are important components of the EO of *B*. *morelensis* [17], this work aimed to determine their wound healing activity.

## 2. Results

### 2.1. Cell Viability and Cell Proliferation Assays

Cell viability tests were performed in monolayer cultures of fibroblasts via live/dead staining, after 24 h of applying PIN and FEL at two concentrations (0.1 and 0.01 mg/mL). Neither concentration of each terpene affected the cell viability, since few red nuclei (dead cells) were observed (Figure 1), corroborating previous results corresponding to *B. morelensis* essential oil that contains these terpenes [17].

Once the cell viability of the fibroblast cultures was estimated, cell proliferation was evaluated by a PrestoBlue assay, where the absorbance is directly proportional to the cell proliferation. The absorbance was found to increase with time. It was evident that a higher concentration of terpenes meant lower fibroblast proliferation (Figure 2). It should be noted that, for all the concentrations of the terpenes, the absorbance significantly increased from 24 h onwards, although in no case did the growth exceed that of the control (DMEM supplemented) (Figure 2).

### 2.2. Wound Healing Efficacy (%WHE)

Because, in the in vitro tests, no adverse effects such as decreased cell viability or alterations in cell morphology were found when applying low concentrations of terpenes, the wound healing activity was tested. The PIN and FEL terpene concentrations used were 9 and 1%, respectively, according to a previous report [10]. Longitudinal wounds on the backs of mice were made to evaluate the WHE. All the wounds had closed after 10 days (Figure 3).

Ten days after the treatments, the WHE percentage was measured by a tensiometric test. The results indicate that the untreated wound (C−) showed 9.18% WHE and the Recoverón NC^®^-treated wound (C+), 13.03%, while the wounds treated with PIN and FEL showed 21.16 and 19.23% WHEs, respectively (Figure 4).

### 2.3. Incision Wound Model

To evaluate the wound contraction and wound closure speed, an incision wound model was used, measuring the diameter and calculating the percentage of closure on days 3, 5, 7, 9, and 10. The results indicate that the wounds treated with PIN and FEL showed the highest wound closing speeds (WCSs) on the third day (Figure 5), where the average percentage of wound contraction per treatment was: C− 0.86%, C+ 4.18%, MO 7.55%, PIN 51.74%, and FEL 25.6%.

After 10 days of treatment, the percentages of wound contraction (WC) were as follows: C−: 61.42%; C+ positive control—Recoverón NC^®^: 57.52%; MO: 52.10%; PIN: 85.69%; and FEL: 85.97%. Significant differences were found between the treatments with EO and terpenes for C−, C+, and MO. Once again, significant differences were found between the PIN and FEL treatments and the control groups (C−, C+ positive control—Recoverón NC^®^, and MO) (Figure 6).

On the tenth day, the wounds presented scabs in the C– control mice, and in the C+ positive control (Recoverón NC^®^-treated) mice. From the third day, the cephalic and caudal sides of the terpene-treated wounds showed a fusiform aspect at the junctions of edges, probably due to the adhesive properties of PIN and FEL. The longitudinal axes of these wounds were perpendicular to the median planes of the mice, and, consequently, the scars that formed had this shape (Figure 7).

The wounds’ appearance was correlated with the histopathological findings after 10 days of treatment. Tissue sections from the healthy skin (HS) of mice, stained with hematoxylin–eosin and Masson’s trichrome method, demonstrated that the skin consists of epidermis, dermis, hypodermis, and panniculus carnosus muscle (Figure 8, HS, H/E, and MT). The epidermis is formed of a keratinized stratified squamous epithelium; the dermis, located immediately below the epidermis, constitutes dense, irregular collagenous connective tissue. The back skin of mice is thin, undulations at the dermal‒epidermal junction are modest, and rete ridges are not observed.

The histopathological analysis of skin three days after a wound demonstrated the presence of scabs, and diffuse, acute inflammatory infiltrates at different levels of the dermis were observed in all the mouse groups. The structure of the skin was very distorted in the positive control group treated with Recoverón NC^®^ (Figure 9, C+) and in the mineral oil (vehicle)-treated mice (Figure 9, MO), with a loss of the stain affinity of connective tissue in the Masson’s trichrome (MT) method (Figure 9, C+ and MO, MT). However, the PIN- and FEL-treated skin still presented some dermal collagen fiber bundles and blood microvasculature vessels inside the mononuclear infiltrate, mainly in the FEL-treated mice (Figure 9, PIN and FEL, H/E and MT). These dermal features suggest the earlier development of immature granulation tissue in the terpene-treated skin than in the negative control, positive control, and mineral oil-treated skin. Moreover, the observed epidermis was thicker in the PIN- and FEL-treated skin than in the healthy skin. Adipocytes from the hypodermis were best conserved in the terpene-treated skin.

## 3. Discussion

Several studies have shown that the EO of *B. morelensis* contains various terpenes, among which α-pinene and α-phellandrene are prominent. These same studies have demonstrated various biological activities of the said EO [13,14,15,17]. In a previous study, we demonstrated that the EO of *B. morelensis* has wound healing activity [17], so in this work, we sought to verify these two terpenes’ wound healing activity at the percentage concentrations in which they were found in said study.

The cell viability test (live/dead, Figure 1) demonstrated that PIN and FEL have low cytotoxicity and that their effects are concentration dependent. The proliferation tests demonstrated that PIN and FEL do not stimulate proliferation in fibroblast cultures (Figure 2). Live/dead viability staining describes dyes where one specifically stains live cells (green fluorescence) while the other dye stains dead cells (red fluorescence); calcein AM with ethidium homodimer-1 involves a membrane-permeable dye which is metabolized within viable cells, mixed with a membrane-impermeable DNA binding molecule. The calcein AM is membrane permeable and is cleaved by esterases in live cells to yield cytoplasmic green fluorescence. The membrane-impermeable ethidium homodimer-1 labels nucleic acids of membrane-compromised cells (i.e., dead) with red fluorescence. The ratio of live to dead cells then can easily be determined by simple counting [29].

Regarding the wound healing activity, our results show that treatment with either terpene led to a better structure in scar tissue and more significant collagen deposition (Figure 8 and Figure 9). Furthermore, the histopathological analysis of the wounds treated for three days showed that, in those where the WHE was improved by both terpenes, slight increases in collagen deposition were observable (Figure 9); this is very important because the wound repair process depends on the biosynthesis, deposition, and maturation of collagen [30]. Likewise, the presence of fibroblasts in the repaired tissues indicates a re-epithelialization process since the fibroblast cells in the reticular dermis migrate into the wound bed and form an extracellular matrix of granulation tissue [31].

It is probable that these collagen deposits in the wounds treated with PIN and FEL provided resistance to tension of the scars formed, since the structure of the skin is designed to minimize the stress on the tissue, so it deforms with movement or external forces. Likewise, it has been described how the different densities of collagen in the papillary dermis, reticular dermis, and hypodermis ensure that the skin can extend in any direction when a force is applied. The skin also contracts in a right-angle plane, with a progressive reduction in volume in the stretched specimen [32].

In addition to the accumulation of collagen, the fabric of the fibers present with the treatments with the two terpenes must have also influenced the WHE, since the collagen network assembly is carefully interlaced so that it can be pulled in any direction. However, there is a favored anisotropy so that more stretching can be applied before a mechanical plateau and endpoint is reached; this is an important feature to understand when trying to close skin wounds, which can adapt to these forces by the mechanical relaxation of collagen fibers and biological remodeling of the fibrous structure, resulting in mechanical and biological “creep” [33].

Collagen deposited by fibroblasts also influences wound contraction and closure speed, since, when tissues break after injury, collagen is needed to repair the defect and restore anatomical structure and function. If too much collagen is deposited at the wound site, the typical anatomical structure is lost, function is compromised, and fibrosis occurs. Conversely, if insufficient collagen is deposited, the wound is weak and can open [34].

Contraction is part of the normal healing process, but it becomes pathological and is known as a contracture if it is excessive. All dermal wounds heal by three elementary mechanisms: the deposition of a connective tissue matrix, contraction, and epithelialization [35]. In the incision wound model, we found that the highest closure speed (Figure 5) and the highest WHE (Figure 4) occurred with the PIN and FEL treatments. Our observations, throughout the application of these treatments, allowed us to notice that the terpenes had an adhesive effect (weak) on the skin, which could generate a healing effect of primary intention on the wounds, for superficial wounds that can be closed by sutures, adhesive tapes, or staples [36].

During a primary intention, the main healing mechanism is the deposition of the connective tissue matrix, where collagen, proteoglycans, and binding proteins are deposited to form a new extracellular matrix. By contrast, wounds that remain open heal primarily by contraction. The interaction between the cells and the matrix causes tissue movement towards the center of the wound [37]; this can also be seen in the histology, since in the three-day PIN and FEL micrographs, very close to the invaginations in the scar tissue, there is healthy tissue that even seems to slide inside the scar (Figure 9).

It is also likely that the terpenes contributed to the wound healing process through antimicrobial activity, since it has been shown that the EO of *B. morelensis*, whose chemical composition includes α-pinene and a-phellandrene, has antibacterial and antifungal activity [14]. More specifically, it is known that PIN alters the expression of the gene encoding the INT1p integrin. This is very important, since it is known that integrins are essential in the adhesion of *C. albicans* [15]. Likewise, α-phellandrene was also shown to inhibit leukocyte rolling and adhesion, the production of the proinflammatory cytokines TNF-α and IL-6, and the degranulation of mast cells induced by compound 48/80; this suggests that α-phellandrene plays an essential role as an anti-inflammatory agent through the modulation of neutrophil migration and the stabilization of mast cells [23].

The small structure of monoterpene molecules facilitates the substances permeating the skin [28]. This effect is probably due to the interaction of compounds with liquid crystals of skin lipids [38]. It is known that terpenes, in comparison to synthetic permeation enhancers, can improve the barrier-crossing activity of both lipophilic and hydrophilic compounds—even in low terpene concentrations [39].

The results obtained in this work agree with those previously published by our group on the healing activity of *B. morelensis*’ EO (which contains the terpenes used here, in the same concentrations). In that work, a probable mechanism of action of *B. morelensis* EO was suggested: promoting fibroblast migration to the wound site, making the fibroblasts active in the production of collagen in the early stages, and, later, promoting the remodeling of this same collagen [17]. The results obtained here show that both PIN and FEL have healing activity on their own. We believe that the mechanisms of action of both are similar to those suggested for *B. morelensis* EO, although with the clear difference that the terpenes act as agents of primary intention in repair wounds, which confers on them the ability to accelerate WCS while resulting in a certain WHE%—that is, a good tensile strength of the scar—without resulting in a contracture [35]. The terpenes PIN and FEL achieve the primary goals of wound treatments, which are rapid closure, together with a functional and aesthetically satisfactory scar [40]. Although our results show primary intention wound healing activity of both terpenes, we consider it important to carry out other studies that allow us to delve into the mechanisms of action of these compounds.

## 4. Materials and Methods

### 4.1. Isolation of Fibroblasts

Fibroblasts were isolated from human skin, obtained by donation with written informed consent. The skin was taken from healthy voluntary donors, using a cylindrical scalpel for 5 mm biopsies, in septic and antiseptic conditions; the skin thus obtained was immediately deposited in Hank solution with an antibiotic. In a laminar flow hood, the skin samples were cut into smaller fragments, and each fragment was grown in Dulbecco’s modified Eagle’s low-glucose medium (DMEM-LG) supplemented with 10% fetal bovine serum (FBS) and antibiotics (100 U/mL penicillin, 100 mg/mL streptomycin, and 100 mg/mL gentamicin), all from Gibco BRL (Rockville, MD, USA), and incubated at 37 °C and in 5% CO_2_. The culture medium was replaced every two days; after two weeks of culture, the explants (skin fragments) were removed. The fibroblasts were cultured to approximately 80% confluence, and the cells were separated with 0.05%/0.02% trypsin/EDTA and reseeded to generate sufficient cells for the subsequent tests [17].

### 4.2. Cell Viability and Proliferation

The cell viability was analyzed in the monolayers of cultured fibroblasts through calcein and ethidium homodimer staining (LIVE/DEAD kit, Thermo Fisher Scientific, Waltham, MA, USA), according to the instructions of the manufacturer. For the assays, 5000 cells/cm^2^ were seeded on glass coverslips coated with poly-L-lysine (Sigma-Aldrich, St. Louis, MO, USA), cultured for 24 h with supplemented DMEM, and stimulated with the terpenes α-pinene (PIN) and α-phellandrene (FEL), both from Sigma-Aldrich. The concentrations of the terpenes were 0.1 mg/mL and 0.01 mg/mL; the terpenes were diluted in cosmetic-grade mineral oil (MO) (Kamecare, Mexico), as was done with the EO of *B. morelensis* in previous work [17]. A death control was obtained by treating cells with ethanol for 30 min before staining. Cells cultured with supplemented DMEM without EO were the control. Panoramic images (200×) were taken using a Nikon Eclipse 80i microscope (Nikon, Shinagawa, Tokyo, Japan) with NIS-Elements F4 (Nikon) software (version Ver4.00.06, Tokyo, Japan). The total number of cells (live and dead) was counted with ImageJ (version 1.52, an open-source Java image-processing program). The viability ratio was calculated according to the following equation:(1)$$viability\;ratio=\frac{live\;cells}{live\;cells+dead\;cells}.$$

For the cell proliferation assay, the cell cultures were incubated with PrestoBlue^®^ reagent (Thermo Fisher Scientific) for 1 h and then the supernatants were placed in 96-well plates. The absorbance of each well’s content was measured at a wavelength of 570 nm using a spectrophotometric plate reader (Thermo Multi Skan Ascent Type 354). Each experiment was conducted three times.

### 4.3. Animals

Male CD-1 strain mice, six to eight weeks old, were obtained from the animal laboratory facility of the FES-Iztacala, UNAM, Edo. Mex., Mexico. The animals, divided into experimental groups consisting of six mice each, were separately housed in ventilated cages under a controlled light cycle (12 h light/12 h dark) at standard room temperature (22–24 °C) and allowed access to a conventional diet and tap water ad libitum. All the guidelines for the care and use of animals were followed (NOM-062-ZOO-1999), and the study was approved by the Institutional Ethics Committee of the UNAM, Facultad de Estudios Superiores Iztacala (CE/FESI/052019/1295).

### 4.4. Wound Healing Efficacy

Mice were assigned to six groups (*n* = 6 mice in each group), and their back skin was shaved. Twenty-four hours later, mice were anesthetized by the inhalation of isoflurane. Aseptic and antiseptic procedures were used for shaved skin, and a 1 cm incision was made. The groups were classified as follows: Group 1: untreated skin without wounds or healthy skin (HS); Group 2: untreated wounds, as a negative control (C−); Group 3: wounds treated with Recoverón NC^®^ (Armstrong Lab, Mexico City, Mexico) (contains: 5% acexamic acid and 0.4% neomycin [41,42]), as a positive control (C+); Group 4: cosmetic-grade mineral oil (Kamecare, Mexico), as a control for the dilution vehicle for the terpenes (MO); Group 6: α-pinene at 9% (Sigma-Aldrich); Group 7: α-phellandrene at 1% (Sigma-Aldrich). The wounds of Groups 4–7 were epicutaneously treated with 10 μL of the respective treatments, whereas the wounds of the control group were covered with Recoverón^®^ cream every 12 h. All the treatments were applied for 10 days. After this time, the mice were sacrificed using a CO_2_ chamber. Immediately after the sacrifice, the wound resistance to tension was measured according to the tensiometric method [43].

The percentage wound healing efficacy was calculated as follows:(2)$$\%\;wound\;healing\;efficacy=\frac{GSS}{GHS}\times 100,\;$$
where GSS is the grams used to open scarred skin and GHS is the grams used to open healthy skin.

### 4.5. Wound Contraction Model

The same groups of mice from the previous experiment were formed to evaluate the wound contraction. In this procedure, a biopsy punch of 5 mm in diameter, not deeper than the hypodermis, was performed, and the same treatment was applied every 12 h for 10 days. Every two days, the wound diameter was measured with a digital caliper (Mitutoyo, Tokyo, Japan), and the percentage of wound contraction was calculated using the following equation:$$\%\;wound\;contraction=100-\frac{wound\;diameter\;on\;specific\;day\;post\;wound\;}{wound\;diameter\;on\;day\;zero}\times 100$$

Day zero corresponds to the day the incisions were made.

### 4.6. Histopathological Observation

On day 10, the animals were sacrificed in a CO_2_ chamber. Skin specimens of wounds were obtained and immediately fixed in 10% buffered formaldehyde over 24 h at room temperature. Afterward, the skin samples were embedded in paraffin to obtain 4 μm thick tissue sections, which were stained with hematoxylin and eosin (H&E) and Masson’s trichrome.

### 4.7. Statistical Analysis

The results are expressed as the means ± standard errors of the means, this is shown in the error bars of all graphs. All the results were subjected to the D’Agostino‒Pearson test for normality, obtaining *P*-values < 0.05. Once the normality of the data was verified, the results were analyzed by employing one-way analysis of variance (ANOVA), with a Tukey‒Kramer multiple comparison post hoc test (*P* < 0.01) using GraphPad Prism 7 software.

## 5. Conclusions

α-pinene and α-phellandrene promote the wound healing process because they generate scars with effective tensile strength, accelerate wound closure, act as an adhesive of primary intention, and contribute to collagen deposition. Our results confirm the healing activity of *B. morelensis* EO, with the aforementioned terpenes explaining part of its demonstrated activity.

## Figures and Tables

**Figure 1 molecules-26-02488-f001:**
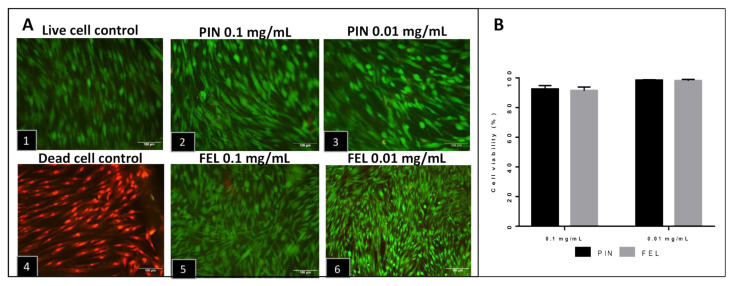
Viability test: Live/dead staining of fibroblast culture 24 h after stimulus with PIN and FEL. (**A1**): Live cell control; (**A2**): α-pinene 0.1 mg/mL; (**A3**): α-pinene 0.01 mg/mL; (**A4**): Dead cell control; (**A5**): α-phellandrene 0.1 mg/mL; (**A6**): α-phellandrene 0.01 mg/mL. (**B**): % Cell viability 24 h after stimulus with PIN (α-pinene, 0.1 or 0.01 mg/mL) or FEL (α-phellandrene, 0.1 or 0.01 mg/mL).

**Figure 2 molecules-26-02488-f002:**
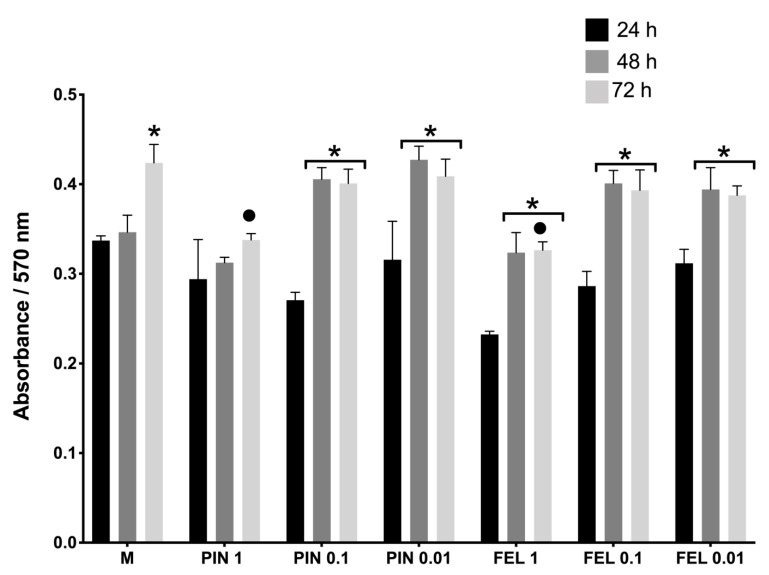
Cell proliferation assay. Resazurin absorbance indicates the proliferation of fibroblasts. Conditions of fibroblast cultures: M = supplemented DMEM, PIN 1 = α-pinene, 1 mg/mL; PIN 0.1 = α-pinene, 0.1 mg/mL; PIN 0.01 = α-pinene, 0.01 mg/mL; FEL 1 = α-phellandrene, 1 mg/mL; FEL 0.1 = α-phellandrene, 0.1 mg/mL; FEL 0.01 = α-phellandrene, 0.01 mg/mL. ***** Significant difference with respect to 24 h of each stimulus. • Significant difference with respect to growth after 72 h in M (growth medium) (*p* < 0.05).

**Figure 3 molecules-26-02488-f003:**
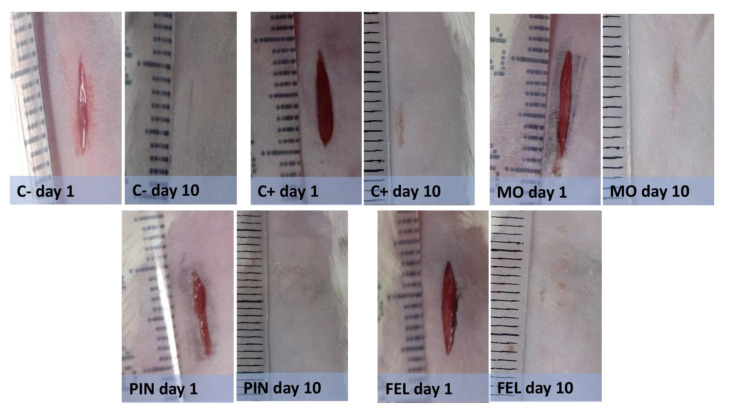
Macroscopic examination on the backs of mice 1 day and 10 day after wounds. C− = untreated wound; C+ = positive control with Recoverón NC^®^ treatment; MO = mineral oil (vehicle); PIN = α-pinene, 9%; FEL = α-phellandrene, 1%.

**Figure 4 molecules-26-02488-f004:**
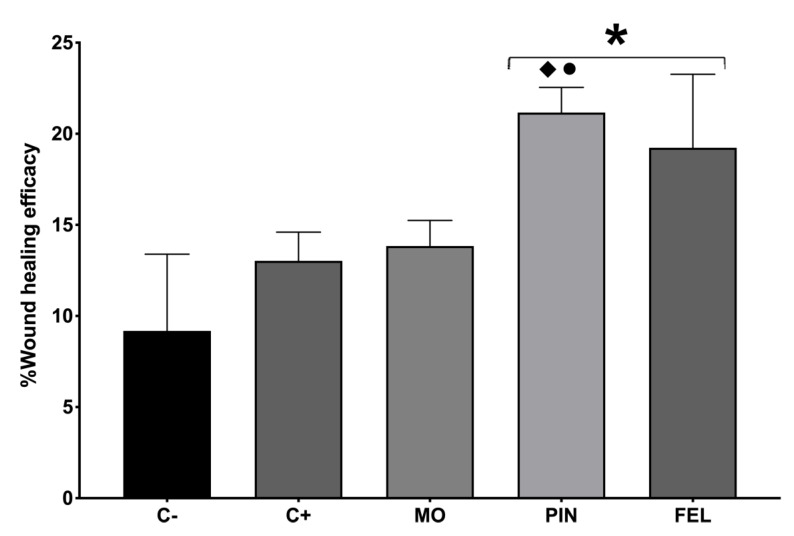
Wound healing efficacy (%WHE). C− = untreated wound. C+ = positive control—Recoverón NC^®^. MO = mineral oil (vehicle); PIN = α-pinene, 9%; FEL = α-phellandrene, 1%. * Significant differences concerning C−. ♦ Significant differences for C+. • Significant differences from MO (*p* < 0.05).

**Figure 5 molecules-26-02488-f005:**
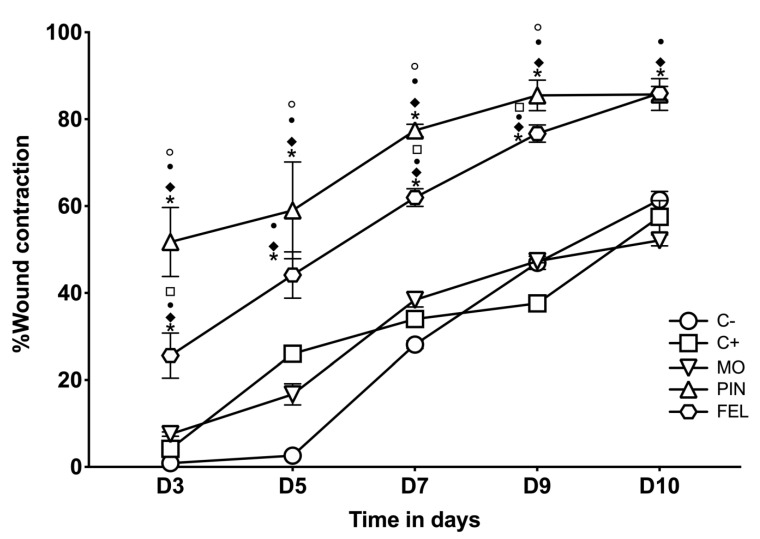
Wound closure speed (WCS)**.** C− = untreated wound. C+ = positive control—Recoverón NC^®^. MO = mineral oil (vehicle); PIN = α-pinene, 9%; FEL = α-phellandrene, 1%. * Significant differences with respect to C−. ♦ Significant differences concerning C+. • Significant differences for MO. ○ Significant differences from FEL. **□** Significant differences concerning PIN (*p* < 0.05).

**Figure 6 molecules-26-02488-f006:**
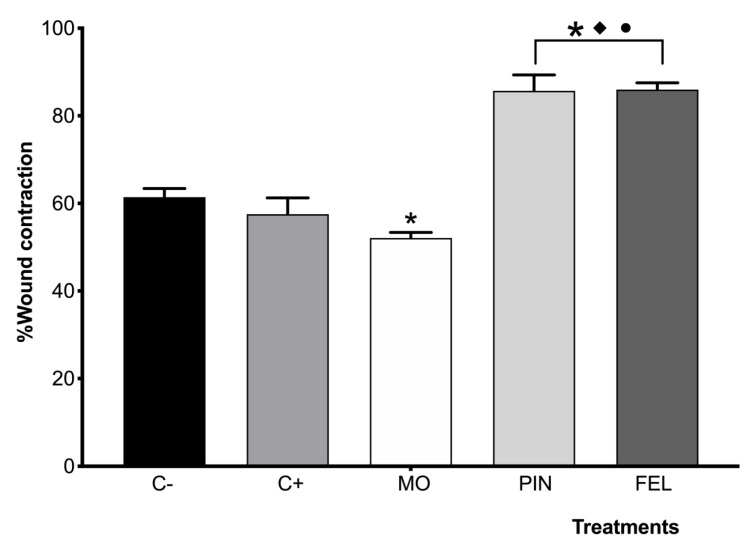
Percentages of wound contraction 10 days after the treatments. C− = untreated wound. C+ = positive control—Recoverón NC^®^. MO = mineral oil (vehicle); PIN = α-pinene, 9%; FEL = α-phellandrene, 1%. * Significant differences concerning C−. ♦ Significant differences for C+. • Significant differences from MO (*p* < 0.05).

**Figure 7 molecules-26-02488-f007:**
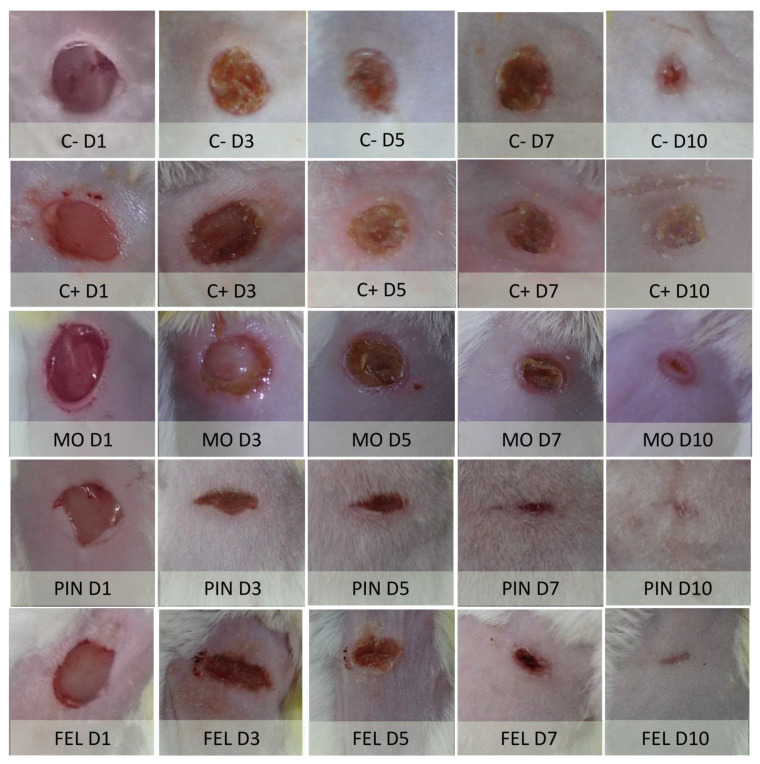
Macroscopic examination of wounds on the backs of mice on the first, fifth, and tenth days of the experiment. C− = untreated wound. C+ = positive control (Recoverón NC^®^ treatment); MO = mineral oil (vehicle); PIN = α-pinene, 9%; FEL = α-phellandrene, 1%.

**Figure 8 molecules-26-02488-f008:**
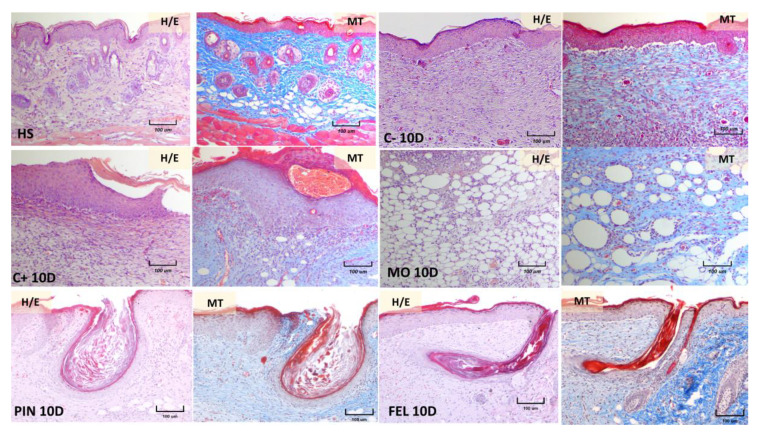
Histopathology of wounds after treatment for 10 days. The skin samples were stained with hematoxylin and eosin (H/E) and Masson’s trichrome (MT). All photos were taken at 10× magnification. HS, healthy skin. C− = untreated wound; C+ = positive control—Recoverón NC^®^; MO = mineral oil (vehicle); PIN = α-pinene, 9%; FEL = α-phellandrene, 1%.

**Figure 9 molecules-26-02488-f009:**
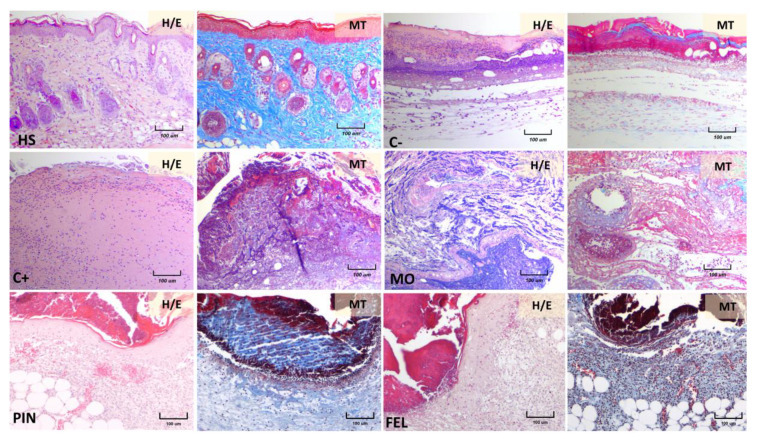
Histopathology of wounds after treatment for three days. The skin samples were stained with hematoxylin and eosin (H/E) and Masson’s trichrome (MT). HS = healthy skin; C− = untreated wound; C+ = positive control—Recoverón NC^®^ treatment; MO = mineral oil (vehicle); PIN = α-pinene, 9%; FEL = α-phellandrene, 1%.

## Data Availability

Data is contained within the article.

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
