# Peer review of "Wound Healing Activity of α-Pinene and α-Phellandrene"

_molecules, 2021, doi:10.3390/molecules26092488_

Round 1
Reviewer 1 Report
In the results, on line 62, they refer to previous results, it is recommended to indicate reference.
Why do the authors use Recoveron as a positive control?
It is recommended to the authors that in figure 1 differentiate with A and B between both images.
In figure 2, Are there significant differences in absorbance between 48 and 72 h in monoterpene treatment?
Line 85 shows Fig. 3 instead of Figure 3.
In figure 3 the negative control of the experiment, untreated wound, does not appear;
In figure 5, it is recommended to change the FEL symbol, as it is similar to the circle.
The results indicate that the fastest wound closure speed,is on day 3, by what percentage? How is it calculated?
In graph 5 the wound contraction (%) is indicated versus time, and in figure 6 the wound contraction is indicated versus the treatment. What are the differences between the two?
The negative control wound contraction values of both graphs do not match, why?
How is the wound closure percentage calculated?
The authors are recommended to go deeper into the discussion of the results.
Why do the authors use ANOVA and the Tukey-kramer Test?
It is recommended to expand the number of bibliographic references, to increase the quality of the article, a maximum of 50.
Author Response
In the results, on line 62, they refer to previous results, it is recommended to indicate reference.
The idea was completed and the previous work was cited: corresponding to Bursera morelensis essential oil that contains these terpenes (Salas-Oropeza et al. 2020)
Why do the authors use Recoveron as a positive control?
The commercial healing medicine Recoverón NC was used as a positive control, which contains: 5% of acexamic acid and 0.4% of neomycin¨. Acexamic acid participates in the protein action of collagen, which allows it to act in the healing process, regulating the production of fibroblasts and the arrangement of collagen fibers within the same natural biological process, but in an orderly manner (Guillard et al. 1987). Neomycin is a broad spectrum antibiotic. Sensitive mechanisms can be inhibited by concentrations of 5 to 10 µg/ml or less (Shaikh et al, 1995).
It is recommended to the authors that in figure 1 differentiate with A and B between both images.
The recommended change has been made
In figure 2, Are there significant differences in absorbance between 48 and 72 h in monoterpene treatment?
No, the significant difference found is within each treatment; that is, in each concentration of monoterpene it was found that there was a significant difference in growth with respect to the 24 hours after the application of the stimulus.
As this absorbance is not greater than that of the monoterpene-free growth medium control, it means that the increase in absorbance is due to a normal behavior of the cells and therefore there is no stimulation of proliferation
Line 85 shows Fig. 3 instead of Figure 3.
The recommended change has been made.
In figure 3 the negative control of the experiment, untreated wound, does not appear.
The recommended change has been made
In figure 5, it is recommended to change the FEL symbol, as it is similar to the circle.
The recommended change has been made
The results indicate that the fastest wound closure speed,is on day 3, by what percentage? How is it calculated?
In the results section the following text was added: where the average percentage of wound contraction per treatment was: C- 0.86%, C+ 4.18%, MO 7.55%, PIN 51.74%, and FEL 25.6%.
In the Methodology section the following text was added to explain how the percentage of wound contraction was calculated:
…using the following equation:
% wound contraction = 100 - wound diameter on specific day post wound/wound diameter on day zero X 100
Day zero corresponds to the day the incisions were made.
In graph 5 the wound contraction (%) is indicated versus time, and in figure 6 the wound contraction is indicated versus the treatment. What are the differences between the two?
Figure 5 shows wound contraction on different days, considering that those that show a higher percentage of contraction in less time are those that are healing faster; while in figure 6 only the wound contraction data found at the end of the treatment application are shown, considering this the final result of the experiment; that is, the total wound contraction of each treatment.
The negative control wound contraction values of both graphs do not match, why?
Both graphs were constructed with the same data for day 10, below I show screenshots where it is observed that in both cases the average at 10 days for C- is 61.42.
https://drive.google.com/drive/u/0/my-drive
How is the wound closure percentage calculated?
In the Methodology section the following text was added to explain how the percentage of wound contraction was calculated:
…using the following equation:
% wound contraction = 100 - wound diameter on specific day post wound/wound diameter on day zero X 100
Day zero corresponds to the day the incisions were made.
The authors are recommended to go deeper into the discussion of the results.
Paragraphs that were added in the discussion are highlighted in yellow.
Why do the authors use ANOVA and the Tukey-kramer Test?
ANOVA was used in the statistical analysis because the data obtained refer to studies of the effect of one or more factors (each with two levels). It is, therefore, the statistical test to use when you want to compare the means of two or more groups.
If an ANOVA is significant, it implies that at least two of the means compared are significantly different from each other. A multiple comparison test of means must be applied to determine which pairs of means differ significantly. The Tukey-Kramer test is the recommended adjustment when the number of groups to be compared greater than 6 and the design is balanced (same number of observations per group).
It is recommended to expand the number of bibliographic references, to increase the quality of the article, a maximum of 50.
18 citations were added to the manuscript
Reviewer 2 Report
The manuscript by Salas-Oropeza et al. describes the wound healing activity of two natural compounds, α-pinene (PIN) and α-14 phellandrene (FEL), which are present in the essential oil of B. morelensis. The authors used in vitro methodology to assess the cytotoxicity of the compounds and in vivo assays to address the wound healing capacity of these terpenes. The manuscript is well written and well organized, the methodology used supports the research question, and the results are of interest for the scientific community. However, some concerns should be addressed:
- Line 35: Several studies have shown that a system of channels in the cortex of Bursera morelensis produces an EO with a high concentration of volatile terpenoids. Please add a reference to support this statement.
- Line 254: “The concentrations of the EO were 0.1 mg/mL and 0.01 mg/mL; the terpenes were diluted in cosmetic-grade mineral oil (MO) (Kamecare, Mexico)”. The authors state the concentration of “EO” when they actually used only PIN or FEL. I believe it would be best to not use “EO” when referring to the terpenes isolated to avoid confusion.
- How did the authors choose the concentration of the monoterpenes for the in vitro assays? For the in vivo assays, PIN was used at 9% and FEL was used at 1% making it difficult to compare the efficacy of both monoterpenes.
- In figure 1 the size of the letters on the graph axis should be increased.
- Figure 7: I believe the last image for the MO group is MO D10, not MO D3. Please confirm.
- The authors are lacking mechanistic analysis to propose the actual mechanism of action of PIN and FEL. More experiments should be provided or the authors should state this as a limitation of the study.
- Are the graphs showing standard deviation or standard deviation of the mean? The authors should include this information in the statistical analysis section or in the caption of every figure.
Author Response
1. Line 35: Several studies have shown that a system of channels in the cortex of Bursera morelensis produces an EO with a high concentration of volatile terpenoids. Please add a reference to support this statement.
References were added to support this information (references 6-11)
2. Line 254: “The concentrations of the EO were 0.1 mg/mL and 0.01 mg/mL; the terpenes were diluted in cosmetic-grade mineral oil (MO) (Kamecare, Mexico)”. The authors state the concentration of “EO” when they actually used only PIN or FEL. I believe it would be best to not use “EO” when referring to the terpenes isolated to avoid confusion.
The recommended change has been made
3. How did the authors choose the concentration of the monoterpenes for the in vitro assays? For the in vivo assays, PIN was used at 9% and FEL was used at 1% making it difficult to compare the efficacy of both monoterpenes.
For the in vitro tests, preliminary tests were carried out with different concentrations of terpenes, determining that at low concentrations no large morphological alterations of the fibroblasts were found, for that reason it was decided to use the concentrations of 0.1 and 0.01 mg / mL for the study.
On the other hand, the concentrations used in the in vivo tests were determined considering the concentration of these terpenes in the essential oil of B. morelensis used in a previous study (Salas-Oropeza et al. 2020). Also in the manuscript the following text was added:
as was done with the EO of B. morelensis in previous work [17] (line 288-289).
4. In figure 1 the size of the letters on the graph axis should be increased.
The recommended change has been made
5. Figure 7: I believe the last image for the MO group is MO D10, not MO D3. Please confirm.
Your observation is correct, the recommended change has been made.
6. The authors are lacking mechanistic analysis to propose the actual mechanism of action of PIN and FEL. More experiments should be provided or the authors should state this as a limitation of the study.
In the discussion section of the manuscript the following text was added:
Although our results show primary intention wound healing activity of both terpenes, we consider it important to carry out other studies that allow us to delve into the mechanisms of action of these compounds. (Line 263-265)
7. Are the graphs showing standard deviation or standard deviation of the mean? The authors should include this information in the statistical analysis section or in the caption of every
The results are expressed as the means ± standard errors of the means. this is shown in the error bars of all graphs. (Line 344-345).